# Insight into Calcium-Binding Motifs of Intrinsically Disordered Proteins

**DOI:** 10.3390/biom11081173

**Published:** 2021-08-09

**Authors:** Estella A. Newcombe, Catarina B. Fernandes, Jeppe E. Lundsgaard, Inna Brakti, Kresten Lindorff-Larsen, Annette E. Langkilde, Karen Skriver, Birthe B. Kragelund

**Affiliations:** 1Linderstrøm-Lang Centre for Protein Science, University of Copenhagen, 2200 Copenhagen, Denmark; estella.newcombe@sund.ku.dk (E.A.N.); catarina.fernandes@bio.ku.dk (C.B.F.); jmd737@alumni.ku.dk (J.E.L.); brakti.inna@gmail.com (I.B.); lindorff@bio.ku.dk (K.L.-L.); KSkriver@bio.ku.dk (K.S.); 2Department of Drug Design and Pharmacology, University of Copenhagen, Universitetsparken 2, 2100 Copenhagen, Denmark; annette.langkilde@sund.ku.dk; 3REPIN, Department of Biology, University of Copenhagen, Ole Maaloes Vej 5, 2200 Copenhagen, Denmark

**Keywords:** intrinsically disordered proteins, IDP, calcium, NMR, SLiM, motifs

## Abstract

Motifs within proteins help us categorize their functions. Intrinsically disordered proteins (IDPs) are rich in short linear motifs, conferring them many different roles. IDPs are also frequently highly charged and, therefore, likely to interact with ions. Canonical calcium-binding motifs, such as the EF-hand, often rely on the formation of stabilizing flanking helices, which are a key characteristic of folded proteins, but are absent in IDPs. In this study, we probe the existence of a calcium-binding motif relevant to IDPs. Upon screening several carefully selected IDPs using NMR spectroscopy supplemented with affinity quantification by colorimetric assays, we found calcium-binding motifs in IDPs which could be categorized into at least two groups—an Excalibur-like motif, sequentially similar to the EF-hand loop, and a condensed-charge motif carrying repetitive negative charges. The motifs show an affinity for calcium typically in the ~100 μM range relevant to regulatory functions and, while calcium binding to the condensed-charge motif had little effect on the overall compaction of the IDP chain, calcium binding to Excalibur-like motifs resulted in changes in compaction. Thus, calcium binding to IDPs may serve various structural and functional roles that have previously been underreported.

## 1. Introduction

Intrinsically disordered proteins (IDPs) are dynamic proteins without a well-defined globular structure, resulting from their lack of hydrophobic amino acids compared to globular proteins. IDPs were originally described as being more charged and polar than globular proteins [1,2,3]; however, it is now clear that IDPs contain a variety of sequences [4], some with minimal charge [5], and others with, for example, enrichment in the hydrophobic proline [6]. While there is diversity in IDP sequences, it holds true that a certain group of IDPs are highly charged, with large excess of positive or negative charges, leading to a large net charge per residue (NCPR). These highly charged IDPs participate in many complexes that rely on their charge density, including interactions with DNA [7,8] as well as with other oppositely highly charged IDPs. One example of IDPs forming a complex reliant on opposite charges is that formed between histone H1 and its chaperone prothymosin α (ProTα). This complex showed, for the first time, that two highly charged IDPs could interact with high affinity, while remaining disordered and dynamic [9,10].

In the cell, several different divalent cations play key biological roles. In DNA binding, both magnesium and zinc ions are coordinated to DNA and DNA-binding proteins, respectively [11,12], and within the cell, as well as outside, calcium ions serve as secondary messengers in most eukaryotic systems. Calcium plays highly versatile roles in signaling, e.g., in cell proliferation and cellular development, and with a particularly important role in neurological signaling in the context of learning and memory [13]. Calcium ions are typically coordinated by six ligands, although variations to this exist with some coordination also made by water molecules [14], suggesting that compared to, e.g., magnesium, calcium coordination can be much more adaptive. The concentration of free calcium fluctuates temporally and spatially from the resting concentration in the cytosol (100 nM) to regionally high concentrations as great as 500 μM [15]. Thus, fluctuation in the calcium concentration has the capacity to turn on and off calcium sensing proteins, and even weak interactions with this divalent cation play relevant biological roles, as in the case of metabotropic glutamate receptors, which have a mM affinity for calcium [16].

The flexibility allowed in calcium binding and the presence of highly negatively charged IDPs, leads to questions regarding if and how IDPs and calcium interact. We know from folded proteins that several protein families are dependent on calcium binding, either to support the structural fold, or for sensing and triggering biological responses, but roles related to calcium binding to IDPs have, to our knowledge, not been systematically explored. Known calcium-binding motifs typically found in folded proteins include the EF-hand, which forms a helix-loop-helix structure upon binding [17]. There are at least 10 different motifs that fit into the EF-hand family, all generally fitting into the loop signature of DXDXDGX_5_E, where X is any amino acid [18]. While IDPs can form a structure upon interaction, some known calcium-binding motifs do not necessarily rely on the formation of α-helices, such as the Excalibur motif. This highly conserved motif was discovered by sequence analysis of extracellular bacterial proteins, identifying the conserved motif DXDXDGXXCE as the calcium binding sequence [19,20]. To achieve interaction at six coordinate points in the Excalibur motif, the Ca^2+^ ion interacts with three negatively charged side chains of aspartic acid in a monodentate fashion. The C-terminal glutamic acid side chain contributes a bidentate interaction, and the carboxyl group of residue seven or a water molecule complete calcium coordination [19,20]. Generally, calcium-binding motifs follow this consensus, with aspartic acid contributing significantly to the coordination of calcium ions.

There are several IDPs that have been shown to interact with calcium; however, they generally do not fit neatly into the canonical EF-hand category or other known motifs. For example, the metal ion-inducible autocleavage (MIIA) domain of a secreted protein from the bacterium *V. coralliilyticus* is disordered and binds calcium at a sequence given by DSDQKGDLEVKE [21]. This clearly follows a similar trend to known motifs, where three aspartic acid residues are involved, along with C-terminal glutamic acid residues; however, the spacing between the negative charges is different to a classical EF-hand and the Excalibur sequence. Another IDP found to bind calcium is Starmaker, which is involved in biomineralization of the zebrafish otolith [22]. Starmaker, despite being highly charged (NCPR: −0.24 [23]), has no identified canonical calcium-binding motif. Interestingly, despite binding >25 calcium ions, Wojtas and colleagues found that although calcium binding altered the hydrodynamic radius of Starmaker, it did not induce structure [24]. This indicates that calcium can interact with disordered proteins without relying on or causing the formation of helices, as observed in the typical EF-hand motif. Furthermore, calcium interaction with the IDP α-synuclein (aSN) has been shown to increase the radius of gyration (*R_g_*) of the protein, effectively increasing the protein size by causing a greater degree of extended conformations, rather than the formation of helices [25]. Thus, it remains a rather open question if calcium-binding motifs exist in IDPs and if so, which roles these would play.

In this study, we aimed to identify whether unique motifs exist for calcium binding within IDPs. We studied the interactions of five carefully selected different IDPs with calcium using nuclear magnetic resonance (NMR) spectroscopy, and designed peptides based on the known calcium-binding IDP, Starmaker, to test the specificity of these interactions. We probed the affinity and stoichiometry of these interactions using colorimetric assays. We also monitored the changes in compactness of the proteins using diffusion NMR. From our data, we identified a calcium-binding motif with properties similar to the Excalibur motif, along with a condensed-charge motif (CCM). Our data suggest that both the flexibility and charge of IDPs are important for their interactions with calcium and that aspartic acid is preferential to glutamic acid in these motifs.

## 2. Materials and Methods

### 2.1. General Protein and Peptide Production

All constructs (proteins sequences Appendix A), except those encoding the C-terminal region of the human sodium proton exchange 1 (NHE1_680-815_), human ProTα, and full-length human aSN (aSN_1-140_), were designed to encode an N-terminal His_6_-SUMO tag, to be cleaved with ubiquitin-like protein protease 1 (ULP1) following initial purification with a nickel column [26]. *E. coli* cells were transformed with one of the following plasmids using heat shock transformation: His_6_-SUMO-aSN_96-140_ (BL21 DE3), His_6_-SUMO-ANAC046_172-338_ (BL21-codon plus DE3; the protein represents the C-terminal region of the *Arabidopsis thaliana* ANA046 protein (ANAC: Arabidopsis no apical meristem [NAM], Arabidopsis transcription activation factor [ATAF], and cup-shaped cotyledon [CUC])), and His_6_-SUMO-DSS1 (BL21 DE3; *Schizosaccharomyces pombe* deleted in split hand/split foot 1). In each case, pre-cultures (10 mL; LB medium kanamycin 50 μg/mL) were used to inoculate M9-minimal medium (1 L) [26,27], containing ^15^NH_4_Cl (1 g/L), ^13^C_6_-glucose (4 g/L), and kanamycin (50 μg/mL). For specific growth times and induction details, see Appendix A. Cells were harvested by centrifugation at 4000× *g* for 10 min, after which the cell pellet was frozen at −20 °C for at least 24 h. Cells were lysed in a lysis/equilibration buffer (Appendix A), using a French pressure cell disrupter (Constant Systems, Daventry, United Kingdom). All His_6_-SUMO proteins were purified initially by immobilized metal affinity chromatography (IMAC), using Nickle Sepharose Fast Flow resin (5 mL; GE Healthcare, Brøndby, Denmark). The column was equilibrated with the same buffer as used in cell lysis, including β-mercaptoethanol (5 mM) for proteins containing cysteine (i.e., ANAC046_172-338_ and DSS1). The lysate was cleared by centrifugation at 20,000× *g*, then filtered through a 0.45 μm filter (Millipore, Darmstadt, Germany). The cleared lysate was incubated with the Nickle Sepharose beads for one hour at room temperature (RT), with gentle shaking. The column was washed with five column volumes (CV) of wash buffer (Appendix A) to remove impurities, followed by 5 × CV equilibration buffer. Finally, the protein was eluted using an elution buffer (Appendix A). Following this initial purification, the His_6_-SUMO tag was cleaved from the protein of interest using the enzyme ULP1 (0.1 mg). The protein was mixed with cleavage buffer (Appendix A) and ULP1, and dialyzed against cleavage buffer, while mixing overnight at 4 °C. The resulting cleavage products were added to the column, equilibrated as above, and in this case the flow through was collected and concentrated using Amicon spin filters (Millipore). Peptides were produced by and purchased from TAG Copenhagen A/S (Copenhagen, Denmark).

### 2.2. aSN Purification

The C-terminal regions of aSN, aSN_96-140_ obtained from the protocol above was further purified using size exclusion chromatography (SEC) with a Superdex 75 10/300 column (GE Healthcare, Brøndby, Denmark) with PBS (137 mM NaCl, 2.7 mM KCl, 10 mM Na_2_HPO_4_, 1.8 mM KH_2_PO_4_). After this, aSN_96-140_ was acidified using trifluoracetic acid (0.1%; TFA) and purified further via reversed phase chromatography (RPC), with the Zorbax C18 column (Agilent, Glostrup, Denmark) on the Äkta Purifier High Performance Liquid Chromatography system (GE Healthcare, Brøndby, Denmark). The protein was eluted using acidified acetonitrile (70% acetonitrile (ACN); 0.1% TFA; (Sigma, Søborg, Denmark)). Following this, aSN_96-140_ was freeze dried and resuspended in HEPES buffer (20 mM HEPES, pH 7.4; Sigma) for NMR analysis. aSN_1-140_ was produced and purified as previously described [28]. NMR spectra were recorded at 5 °C.

### 2.3. ANAC046_172-338_ Purification

ANAC046_172-338_ was further purified by acid precipitation using TFA (1% final concentration), followed by centrifugation at 20,000× *g* for 10 min as described [26]. Precipitated ANAC046_172-338_ was resuspended in resuspension buffer (100 mM Na_2_HPO_4_/NaH_2_PO_4_, pH 8.0; Sigma). This step was repeated twice, and followed by SEC, using running buffer containing DTT (20 mM Na_2_HPO_4_/NaH_2_PO_4_, 100 mM NaCl, 1 mM DTT, pH 7.0). Following purification, the buffer was exchanged to HEPES buffer as above for NMR analysis. NMR spectra were recorded at 10 °C.

### 2.4. NHE1_680-815_ Production and Purification

NHE1_680-815_ was produced using M9 medium as previously described [29,30]. NHE1_680-815_ was exchanged into HEPES buffer as above using Amicon spin filters (Millipore, Darmstadt, Germany). NMR spectra were recorded at 10 °C.

### 2.5. DSS1 Purification

All DSS1 variants were purified using a Source 15 RPC ST 4.6/100 column (GE Healthcare), equilibrated with RPC binding buffer (50 mM NH_3_HCO_3_, 5 mM β-mercaptoethanol). Proteins were eluted using a linear gradient of 0–40% elution buffer (70% ACN, 50 mM NH_4_HCO_3_, 5 mM β-mercaptoethanol). Following purification, DSS1 and variants were lyophilized and resuspended into HEPES buffer as above. NMR spectra were recorded at 10 °C.

### 2.6. ProTα Purification

Human ProTα was purified as previously described [31]. Once pure, the buffer was exchanged (50 mM Tris pH 7.4, 100 mM NaCl) and NMR was measured at 10 °C.

### 2.7. NMR Assignments

Backbone NMR assignments were available for ProTα (BMRB 27215), NHE1_680-815_ (BMRB 27812), DSS1 (BMRB 27618), and aSN [28]. Assignments of ANAC046_172-338_ (BMRB 51033) and DSS1 variants were done from a series of HNCA, HNCOCA, HNCO, HNCACO, CBCACONH, and HNN 3D spectra, obtained essentially as described for WT DSS1 [32]. Assignments of peptides (produced commercially by TAG Copenhagen A/S, Copenhagen, Denmark) were established from spectra recorded on a Bruker 800 MHz spectrometer equipped with a cryogenic probe and Z-field gradient using natural isotope abundance (peptide concentration 1.8 mM, 20 mM HEPES; pH 7.4), acquiring TOCSY, ROESY, ^15^N-HSQC, and ^13^C-HSQC for assignment. Spectra were obtained via Bruker Topspin v3.6.2, transformed using qMDD [33] and processed through NMRDraw (of NMRPipe [34]), before being analyzed manually in CCPN Analysis v2.5 [35].

### 2.8. NMR Data Acquisition and Analysis

All NMR spectra were recorded on Bruker 600 or 800 MHz spectrometers equipped with a cryogenic probe and Z-field gradient using Bruker Topspin v4.0.7, recording titrations at a protein concentration of 50 μM (for peptides 1.8 mM), with CaCl_2_ at a concentration of 2.5 mM. Control for baseline shifts in the absence of calcium was performed with addition of EDTA (2.5 mM). Combined amide (N, H^N^) chemical shift perturbation (CSPs) induced by CaCl_2_ were measured by CCP-analysis version 2.5.

### 2.9. O-Cresolphthalein Complexone Measurements

O-cresolphthalein complexone (OCPC) allows colorimetric readout for free calcium in solution [36]. A stock solution of OCPC (Sigma) was made in deionized water (10 mg/mL, pH 7.6). The reaction was made using OCPC stock (10 μL), Tris-HCl (final concentration 15 mM, pH 8.0), and CaCl_2_ standard (10 μL of 50–500 μg/mL stock solutions), with a final volume of 75 μL in a 96-well plate. Absorption of OCPC was measured at 575 nm on a plate reader (Tecan Infinite Nano, Männedorf, Switzerland). Following this, Starmaker (SM) peptides (400 μM) were added to a concentration of CaCl_2_ (300 μM). Proteins (100 μM) were added to CaCl_2_ (600 μM). Standard curves were fit using a linear fit (Graphpad Prism), and used to calculate concentrations of free and bound calcium. To create Scatchard plots, CaCl_2_ (60–600 μM) was titrated into fixed concentrations of proteins (100 μM; for ProTα: 10 μM) or peptides (200 μM), and free calcium was determined using OCPC absorbance as above. We fitted the data using a simple linear regression, processed by GraphPad prism. From the Scatchard plots, *K*_d_ was calculated as:(1)$$K_{d}=-\frac{1}{slope}$$

Stoichiometry was calculated as follows:(2)$$n=\frac{Bmax}{\left[protein\right]}$$
where *n* is the number of ligands and *Bmax* is the calcium concentration at the *x*-intercept.

### 2.10. Diffusion Ordered NMR Spectroscopy

Translational diffusion constants for each protein (50 μM) and 1,4-dioxane (0.2% *v*/*v*; used as an internal reference) were determined by fitting peak intensity decays within the methyl region centered around 0.85-0.9 ppm from diffusion ordered spectroscopy (DOSY) experiments [37], using the Stejskal-Tanner equation as described [38]. Spectra (a total of 32 scans) were recorded on a Bruker 600 MHz equipped with a cryoprobe and Z-field gradient, and were obtained over gradient strengths from 2 to 98% with a diffusion time (*Δ*) of 200 ms and gradient length (*δ*) of 3 ms in the presence (12.5 mM) and absence of CaCl_2_. aSN_1-140_ (100 μM) was measured with a diffusion time (Δ) of 200 ms and gradient length (*δ*) of 2 ms in the presence (25 mM) and absence of CaCl_2_. Diffusion constants were fitted in Dynamics Center v2.5.6 (Bruker). Diffusion constants were used to estimate the hydrodynamic radius (*R_h_*) for each protein as described [38].

### 2.11. Sequence Properties

For all proteins and peptides, sequence properties including net charge per residue (NCPR), κ-values, hydropathy, and amino acid content were calculated using CIDER [23], and pI was obtained from the ExPASy server [39].

## 3. Results

### 3.1. Negatively Charged IDPs Interact with Calcium in the Mid-Micromolar Range

To probe for calcium binding, we selected five negatively charged IDPs (aSN_96-140_ [40], ANAC046_172-338_ (Appendix A), NHE1_680-815_ [30], DSS1wt [32], and ProTα [41]) from our portfolio of IDPs. The selection was based on a diversity within the following features: the NCPR of these IDPs range between −0.07 and −0.40 (Table 1), with κ-values within the range of 0.17 to 0.42. This value, ranging 0 ≤ κ ≤ 1 corresponds to the patterning of charges, where a low value indicates that negatively and positively charged amino acids are well-mixed and dispersed throughout the sequence, while a higher value indicates that charges are more clustered. Thus, the chosen proteins are all different in size and charge distribution, making them good candidates for identifying features essential for calcium binding. aSN_96-140_ served as a positive control.

To identify the regions within the proteins interacting with calcium, we measured the chemical shift perturbations (CSPs) via NMR at a calcium concentration of 2.5 mM and with protein concentrations of 50 μM (Appendix A). We found regions within each IDP in which CSPs were elevated above the noise (Figure 1A–E), and no changes were seen in the presence of EDTA (2.5 mM; Figure 1F). First, we studied aSN_96-140_, which has previously been reported to interact with calcium [42]. Here, we observed a similar binding pattern as described previously, with the highest CSPs occurring between residues D121 and E137 (Figure 1A). To probe the calcium binding further, we used the OCPC assay to determine the amount of calcium bound by 100 μM protein (Figure 1G). OCPC changes color based on the amount of free calcium in solution, and after obtaining a standard curve based on the protocol by Connerty and Briggs (1966) [36], we could determine the amount of free calcium present in the sample using absorbance. For aSN_96-140_, we found that 100 μM aSN_96-140_ bound approximately 100 μM calcium. Since ionic strength impacts the amplitude of the chemical shift changes of the charged residues, and thus interferes with obtaining a reliable *K*_d_ from NMR titration data, we used the OCPC assay to extract *K*_d_. We titrated each protein with increasing concentrations of CaCl_2_ and measured the OCPC absorbance. From these measurements, the fraction of both bound and free calcium could be extracted, which jointly allowed for affinity determination from Scatchard plot analyses (Appendix A). By fitting the data with a simple linear regression, we found that the *K*_d_ value for the interaction of aSN_96-140_ with calcium was ~50 μM (Figure 1H), similar to what had been previously reported (21 μM) [42]. This assay also gave us information about the stoichiometry of the interaction, which for aSN_96-140_ was ~1:1. This is lower than expected from the literature; however, higher-order complexes of calcium and protein appear transient and may not be observable using the OCPC assay, in contrast to the MS data of the previous study [42].

We next assessed the interaction between calcium and the transcriptional activation domain of the plant transcription factor ANAC046_172-338_ (Figure 1B), which had lower overall NCPR, but a moderate κ-value. Here, based on the CSPs, we observed binding in the region of M315 to S338, indicating a localized C-terminal interaction occurring at the same binding site at which ANAC046 interacts with radical induced cell death1 (RCD1) [43]. We found that ANAC046_172-338_ also bound calcium at a 1:1 ratio (Figure 1G,I) with a *K*_d_ of ~120 μM (Figure 1H). Subsequently, we used the C-terminal disordered tail of the sodium-proton exchanger 1 (NHE1_680-815_) to address the interaction between calcium and a protein with high negative charge at the center of the sequence, finding that this was indeed the region (S745–S766) with the highest CSPs (Figure 1C). NHE1_680-815_ similarly bound calcium at a 1:1 ratio (Figure 1G,I), again with micromolar affinity (~130 μM; Figure 1H). We included the highly multispecific yeast protein DSS1wt, which has a high κ-value, indicating regions of repetitive charged residues of the same charge. In this case, we found two regions of higher CSPs, one between residues S8 to D14, and one between residues T39 to N43 (Figure 1D). Although these regions had the highest CSP, we observed generally high CSPs throughout the protein, i.e., S8–D14, D18–E32, and T39–N43. This is likely because DSS1wt binds calcium at a higher ratio of 1:4 (Figure 1G,I), which may cause overlap within the regions with high CSPs. The overall affinity of DSS1wt for calcium was measured to be ~70 μM (Figure 1H). Furthermore, we also investigated the calcium binding capacity of the highly negatively charged ProTα (Figure 1E), which has been shown to interact with both calcium and zinc [44]. We found ProTα to interact with calcium at micromolar affinity (~70 μM; Figure 1H), which was similar to and only slightly higher than the affinity reported by Chichknova and colleagues (~135 μM) [44]. The NMR peaks from the highly acidic stretch overlap and addition of calcium led to large changes in chemical shifts, meaning that we could not follow all individual peaks and quantify their CSPs (regions indicated by red in Figure 1E). Similar to DSS1wt, ProTα has a high κ-value, as well as a high NCPR. This is reflected in its affinity for calcium and a greater number of calcium binding sites, with ProTα interacting with ~10 calcium ions (Figure 1I), the same stoichiometry as was previously reported [44].

In conclusion, all negatively charged IDPs tested here bound Ca^2+^, either in very local areas as aSN_96-140_, ANAC046_172-338_, and NHE1_680-815_, or with a more distributed binding profile and higher stoichiometry as seen in DSS1wt and ProTα. We suggest that the properties (i.e., the NCPR, κ, and hydropathy; Table 1) of each protein likely contributes to its binding profile. It appears that the NCPR has the greatest influence on the *K*_d_; however, κ-values are also important, suggesting that a regional charge density may overcome the global NCPR. For all proteins, affinities in the ~50–100 μM range were observed.

### 3.2. Peptides from the Starmaker Sequence Reveal the Importance of Negative Charge Distribution in Calcium Binding

To further identify sequence-specific information regarding calcium and IDP interactions, we focused on Starmaker, a calcium-binding IDP essential for the formation of the otolith within the zebrafish ear found to coordinate 28 calcium ions [24]. We, therefore, acquired four peptides (SM1-SM4), taking sequences directly from the Starmaker protein, chosen based on the distribution and number of negatively charged amino acids (Table 2). The peptide sequences ranged from few negative charges distributed across the sequence to many negative charges, with most sequences chosen to include a combination of negatively charged and hydrophobic residues. SM4 was selected due to the repetitive aspartic acid-serine sequence, which is present throughout Starmaker. We assigned the NMR signals of the peptides and measured CaCl_2_-induced CSPs at a 1:1 mix (Appendix A), as well as binding capacity via the OCPC colorimetric assay. CaCl_2_ induced minimal CSPs in SM1, but caused clear CSPs in both the SM2 and SM3 peptides, indicating interaction with calcium, despite having fewer negatively charged residues than SM4 (Figure 2). For SM4, we observed a general broad distribution of the CSPs indicating either calcium binding throughout the peptide or inductive effects from increased ionic strength (Figure 2D). To test the latter, we used the OCPC assay to quantify the amount of calcium bound to the Starmaker peptides. We found that at 400 μM, both SM2 and SM3 bound 200 and 250 μM calcium, respectively, while SM1 and SM4 bound significantly less (Figure 2E). Furthermore, by titrating CaCl_2_ into the SM peptides (200 μM), we were able to calculate the *K*_d_ and stoichiometry of the interactions using Scatchard plots (Appendix A). We found that SM interactions with calcium occurred within the range of 50–80 μM (Figure 2F), and that SM2 and SM3 both bound calcium at a 1:1 ratio, while SM1 bound at a ratio of 1:0.5 (Figure 2G). The titration data for SM4 could not be fitted to calculate *K*_d_ or stoichiometry (Figure 2G). This indicates that two SM1 peptides may be required for an interaction with calcium and may hint that calcium induced oligomerization occurs. This may also account for the minimal CSPs caused by CaCl_2_ in the case of SM1. At a higher NCPR, as seen with SM4, CSPs do not necessarily reflect interaction, but rather the increased effect of ionic strength of the solution on the chemical shifts. It is, therefore, important to supplement NMR data with another assay, such as the OCPC assay, to be able to differentiate the effects of ionic strength from the interaction with calcium. These data gave important clues regarding what was required of an IDP to interact with calcium, i.e., the necessity of negative charges, a κ-value over 0.1, and sufficient hydropathy in combination.

### 3.3. The Significance of Aspartic Acid in Calcium-Binding IDPs

We next analyzed the sequences in the proteins and the peptides with confirmed Ca^2+^ binding and highest CSPs and categorized them into two groups based on sequence features: an Excalibur-like motif and a condensed-charge motif (CCM) (Figure 3A). We observed similar features to the Excalibur-motif [20] in the first group, with two to three aspartic acid residues, followed by a glutamic acid. The CCM reflects regions with a high density of negative charge, without any specific order. It is possible that some of the sequences within the CCM group also act as an Excalibur-like motif, based on the distribution of aspartic acid within the negatively-charged cluster. Additionally, as mentioned above with respect to the SM peptides, we observed that calcium binding did not occur in the negatively-charged SM4, which has no hydrophobic residues. There does not seem to be an exact consistency in the spacing of aspartic acid residues, indicating that the distance between negative charges can vary more in IDPs, than in proteins which, e.g., exhibit the canonical EF-hand.

We noticed an apparent preference for aspartic acid over glutamic acid, so we used DSS1 to test this observation. For this, we produced variants of DSS1 in which all negative charges originated from either glutamic (DSS1E) or aspartic acid (DSS1D), and a fourth variant in which all negatively charged residues were swapped from glutamic to aspartic acid and vice versa (DSS1swap) (Figure 3B). We measured calcium binding capabilities of each mutant using the OCPC assay, finding that DSS1wt, DSS1swap, and DSS1D all bound calcium to a similar extent (Figure 3C). This capacity for calcium binding was reduced four-fold in the DSS1E mutant, indicating that there is a clear preference for aspartic acid within calcium-binding motifs. We further probed the interaction between each variant and calcium using the OCPC assay and derived Scatchard plots (Appendix A) to extract the *K*_d_ for each protein as well as stoichiometry (Figure 3D). From this we found that both DSS1wt and DSS1D bound calcium with comparable affinities, whereas the DSS1swap lost some affinity for calcium. We were not able to properly fit the data for DSS1E as it bound little calcium in this assay. DSS1wt, DSS1swap, and DSS1D each bound three to four calcium ions, and again, we could not fit the data for DSS1E (Figure 3E). The absence of glutamic acid does not detrimentally alter the ability of DSS1 to interact with calcium, and we suggest that this is due to aspartic acid coordination being essential for calcium binding. In contrast, when all negative charges originated from glutamic acid, bound calcium was reduced four-fold and the *K*_d_ could not be accurately determined (Figure 3C,D). Furthermore, introducing glutamic acid in place of aspartic acid led to lower affinity for calcium (Figure 3D). Thus, while some binding may still occur via glutamic acid alone, aspartic acid is clearly preferable.

### 3.4. Conformational Changes Induced by Calcium Interaction

We next aimed to identify whether the addition of calcium to IDPs alters their global conformation in any way. Here, we measured the radius of hydration (*R_h_*) using the internal standard 1,4-dioxane. Full-length aSN_1-140_ became slightly more extended upon calcium addition (Figure 4A), which fits with previous research demonstrating that long-range N- and C-terminal interactions within aSN are broken upon calcium interaction [25]. We saw a small, 1.5 Å reduction in *R_h_* upon calcium binding to ANAC046_172-338_ (Figure 4B), which agrees with the identified site of interaction at the C-terminus. Furthermore, we found that the *R_h_* of NHE1_680-815_ reduced more severely by ~8 Å (25.6%) upon calcium addition (Figure 4C), again related to the position of the binding site at mid-sequence. These data suggest that in some cases, the position of the Ca^2+^-binding site can affect the extension of the IDP by inducing a conformational shift, seen here for both NHE1_680-815_ and ANAC046_172-338_, or through elongation as seen with aSN_1-140_ (Figure 4D). We next acquired DOSY spectra for DSS1wt and its mutants, finding that both DSS1wt and DSS1D became more compact upon calcium binding (25.7% and 17.6%, respectively), while the DSS1swap was unchanged (Figure 4E–G). The *R_h_* of DSS1E was 10.5% smaller in the presence of calcium (Figure 4H), indicating that, while the affinity is slightly lower, its conformation is still affected by calcium binding. It is possible that the differences in DSS1 *R_h_* may be due to the type of interaction occurring, with Excalibur-like (which are more similar to the canonical EF-hand) inducing compactness within the protein, seen for DSS1wt and DSS1D, whereas binding to a CCM results in less structural change (DSS1swap and DSS1E).

## 4. Discussion

In this study, we aimed to identify whether IDPs fit into the canonical calcium-binding motifs reported for globular proteins. We found that IDP calcium binding was similar to the Excalibur motif; however, the spacing between charges was not as consistent, leading us to suggest it to be Excalibur-like. In addition, we found that condensed-charge density also contributed to calcium binding, naming this a condensed-charge motif (CCM). In this case, calcium may bind in a dynamic manner, with the high charge density causing an overall attraction between the negatively charged protein and positively charged calcium, without the formation of a fixed coordination.

We initially showed that five different IDPs interacted with calcium with a moderate affinity, with *K*_d_ values in the ~100 μM range and with various stoichiometries. Calcium-binding proteins can have a wide range of affinities due to their different roles in calcium signaling pathways [14]; however, there have been surprisingly few reports of calcium-binding IDPs, despite their likelihood of high charge density [1]. The largest group of calcium-binding IDPs are involved in biomineralization [45,46], and include Starmaker [24], otolith matrix macromolecule-64 [47], and the casein family [48]. Calcium affinities observed for other IDPs have been within the μM to mM range [21,24], so it is possible that the moderate to low affinity for calcium observed for IDPs, and the technical difficulties in quantifying them, contribute to the reason they have not been uncovered to a greater extent.

The differences that we found in affinities may be due to the differences in NCPR, κ-values and hydropathy (Table 1 and Table 2). We found that the NCPR and κ-values were generally lower (i.e., more negative) and higher, respectively, for proteins binding calcium at a higher affinity. These values both reflect different features regarding the charge of the protein, that is, NCPR reflects the overall charge of the protein, while the κ-value denotes the presence of charged zones throughout a protein. Hence, proteins or peptides with a NCPR closer to zero (e.g., ANAC046_172-338_), but higher κ-value, are still able to bind calcium at a similar affinity to proteins with a more negative NCPR (e.g., aSN_96-140_). These values also provide a case for the importance of hydrophobic residues within the sequence, with calcium binding observed only in proteins and peptides having a hydropathy value of ~2.5 or above. This seems to have affected the calcium binding capability of the SM4 peptide, which consists of a repetitive aspartic acid serine sequence. This was the region of the protein that had been hypothesized to interact strongly with calcium [24]; however, the lack of hydrophobic residues may be the cause of the reduced interaction. It is also likely that phosphorylation of serine would increase calcium binding within this region, as phosphorylated Starmaker is known to interact with a greater number of calcium ions [24]. From the sequences, it is also apparent that more hydrophobic residues often reside in the motif, suggesting that these can act as push buttons to stabilize the structure around the calcium ions, and do this in such a way that it enables the redistribution of the ensemble as well as compaction or expansion, as seen for all the proteins exploiting Excalibur-like sequences (Figure 4). Another motif associated with calcium interaction is the repeat-in-toxin motif, which consists of the sequence GGXGXDXΦX, where X is any amino acid and Φ indicates a large hydrophobic residue [45,49]. It is, therefore, likely that hydrophobic residues are also important for ensuring appropriate spacing between negatively charged residue. We hypothesize that hydrophobic residues are important for calcium interactions, as they create a stabilizing effect via increased entropy from the release of bound water, although further studies are needed to fully decompose the exact role of hydrophobicity.

We also observed that aspartic acid was preferred in calcium-binding proteins over glutamic acid. We tested this by replacing all negative charges of DSS1 with glutamic acid, and thereby, we quartered the amount of calcium it could bind (Figure 3C). This was consistent through the majority of proteins tested in this study, in addition to the many variations on the EF-hand motif [18]. This is likely due to the size of aspartic acid compared to glutamic acid. The ion radius of Ca^2+^ is 0.99 Å [50], leading to the capacity for single coordination points with three aspartic acid residues within close proximity, followed by a bidentate interaction with glutamic acid [51], as seen in EF-hand motifs. Thus, it is likely that the Excalibur-like sequences observed here also bind in this fashion, despite having less sequence homology between proteins. We saw a range of different structural effects caused by calcium interaction, seemingly dependent on the region of interaction within a protein, and possibly linked to the type of calcium-binding motif. This reflects the diversity of IDPs and confers what has been described previously, with some IDPs becoming structured upon calcium interaction, and some remaining disordered [45].

With respect to the CCM, we hypothesize that these interaction sites remain dynamic with calcium bound with in a fuzzy complex. Fuzzy complexes have recently been defined as a ligand having two or more binding sites for a receptor and vice versa [52]. We suggest that this may be the case for proteins with a CCM and calcium, with six coordination points on calcium, and a high overall charge created by several sequential negatively charged amino acids. We observe this especially for ProTα, which was able to remove substantial amounts of calcium from solution (Figure 1G), despite the interactions having relatively low affinities. ProTα is known to participate in highly charged, dynamic interactions [9], as well as interact with other divalent ions (particularly Zn^2+^) at the polyglutamate stretch [53], which contributes to the CCM described here. Chichkova and colleagues reported that ProTα interacted with 10 Ca^2+^ ions and 13 Zn^2+^ ions, with *K*_d_ in the micromolar range for each divalent cation [44]. We suggest that the Excalibur-like sequence may be specific for Ca^2+^, whereas the CCM lacks specificity and is more likely to bind other divalent cations based on charge alone.

We did not, in our study of the proteins, observe the formation of higher-order oligomers at population sizes that could manifest in larger average *R_h_*-values or significant NMR line broadening. However, other ions, such as Cu^2+^ and Zn^2+^, have been shown to lead to transient trimers [54] or dimers and oligomers [55], respectively. One reason for these differences may be the less strict coordination chemistry needed for Ca^2+^ compared to Zn^2+^ and Cu^2+^ [56], which otherwise may be satisfied by oligomer formation. It is likely that the SM1 peptide, which had a stoichiometry of 1:0.5 with calcium may oligomerize to undergo binding. In the context of the Starmaker protein, this may explain one way in which the repetitive serine-aspartic acid regions contribute to coordination of calcium.

Here, we have analyzed calcium binding to the IDPs selected based on sequence and structure features and we found calcium binding across a diverse functional repertoire. It is, therefore, interesting to speculate on the biological role of calcium binding to IDPs and the possible differences between Excalibur-like and CCMs. It is important for the cell to scavenge calcium ions and prevent cytoplasmic calcium-phosphate precipitation [57], although matrices of calcium-phosphate precipitates exist in mitochondria [58]. By binding calcium with moderate affinities, negatively charged IDPs with CCMs may jointly contribute to keeping the free calcium concentration low—but available—via disordered-buffering capacities. Calcium as a secondary messenger may also induce structural changes via Excalibur-like motifs in IDPs, similar to its effect on folded proteins. In the case of calcium, affinity does not always confer biological relevance, as many low-affinity calcium interactions are biologically relevant, including interactions with glutamate receptors [16] and other transmembrane proteins [59]. With the affinities in the 50–100 μM range uncovered in the present work, as well as in previous studies [42,44], it may very well be likely that calcium binding to IDPs may serve regulatory functional roles, which, however, remain to be uncovered. IDPs often engage in multivalent interactions [60], including those leading to liquid-liquid phase separation [61,62,63]. Emerging research shows that phase separation can be regulated by calcium in the endoplasmic/sarcoplasmic reticula to enable efficient calcium buffering [64]. Taken together, this suggests that the negatively charged IDPs may work to sponge calcium towards the multi-protein complexes, thus regulating calcium bioavailability and signaling.

## 5. Conclusions

In this study, we have identified new calcium-binding IDPs (NHE1, ANAC046, and DSS1), and analyzed their sequences along with some known calcium-binding IDPs. We have shown that these IDPs bind calcium using sequences similar to the Excalibur domain, which causes differential structural changes dependent on the IDP as well as using a condensed-charge motif interacting with calcium via a “fuzzy” mechanism relying primarily on charge. Binding affinities were in all cases in the ~50–100 μM range, suggesting that IDPs bind calcium to serve regulatory roles. How binding of calcium to IDPs contributes to function besides those of biomineralization remains an open question.

## Figures and Tables

**Figure 1 biomolecules-11-01173-f001:**
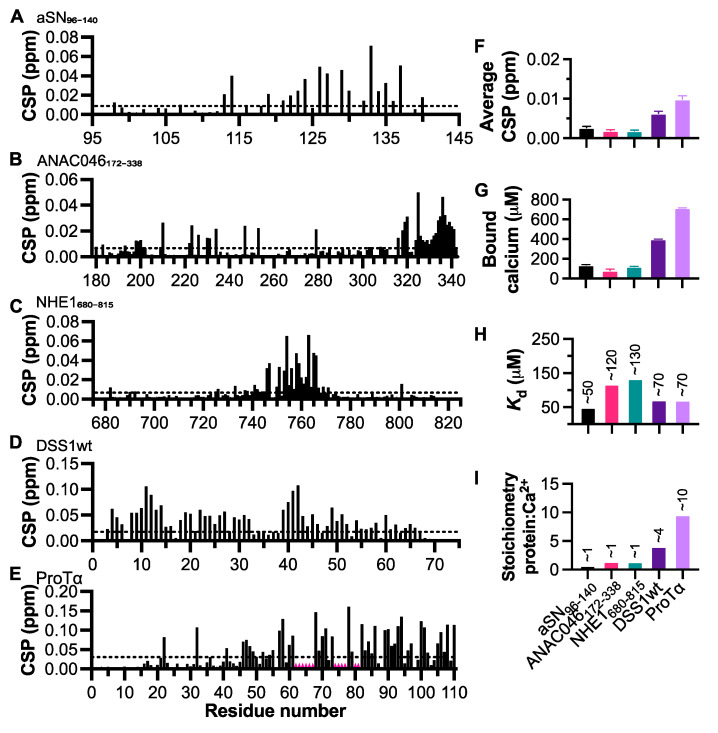
Calcium binding measured by NMR and OCPC absorption. Combined amide (N, H^N^) CSPs observed in different IDPs (50 μM) following addition of CaCl_2_ (2.5 mM; Appendix A). (**A**) CSPs caused by the addition of CaCl_2_ (2.5 mM) to aSN_96-140_, (**B**)ANAC046_172-338_, (**C**) NHE1_680-815_, (**D**) DSS1wt, and (**E**) ProTα. (**F**) Averaged CSPs induced by EDTA (2.5 mM). (**G**) Concentration of calcium bound by the above proteins (100 μM), as measured by the OCPC assay. (**H**) *K*_d_ and (**I**) stoichiometry as calculated from Scatchard plots for each protein (see Appendix A). Error bars indicate SEM.

**Figure 2 biomolecules-11-01173-f002:**
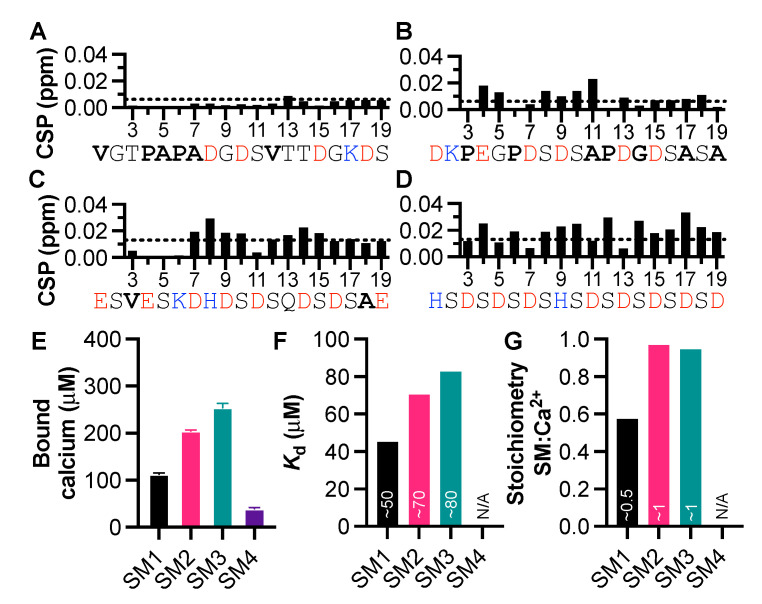
Narrowing down Ca-binding motifs using Starmaker peptides. (**A**) Amide CSPs caused by treatment of SM peptides with 1:1 CaCl_2_: (**A**) SM1, (**B**)SM2, (**C**) SM3, and (**D**) SM4. The dotted line indicates the threshold for CSP intensity, red indicates negative charge, blue indicates positive charge, and bold indicates hydrophobic residues. (**E**) Amount of calcium bound by each SM peptide (400 μM), as measured using the OCPC assay. Error bars indicate SEM. (**F**) *K*_d_ and (**G**) stoichiometry of the interaction between SM peptides and Ca^2+^ calculated using Scatchard plots (Appendix A).

**Figure 3 biomolecules-11-01173-f003:**
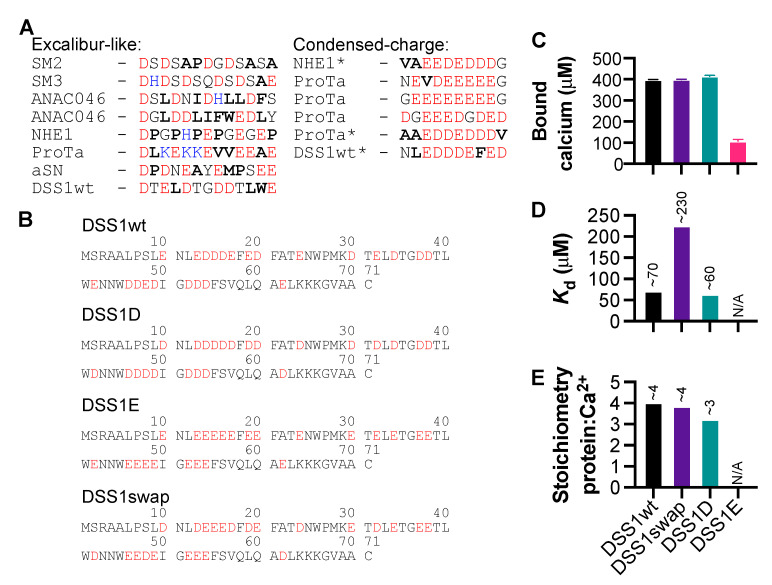
Narrowing down important calcium binding features in IDPs. (**A**) Two types of motifs were often associated with CSPs: the Excalibur-like sequence, involving aspartic acids distributed amongst other amino acids, in particular hydrophobic amino acids (bold); and a condensed-charge motif, which has repetitive negatively charged amino acids createing an overall high density of charge. * Indicates sequences which may fall into either category based on charge distribution. (**B**) DSS1 variant sequences. (**C**) Amount of calcium bound by each of the DSS1 mutants (100 μM). (**D**) *K*_d_ of the interaction between each protein and Ca^2+^. (**E**) Stoichiometry of the interaction between each protein and Ca^2+^. Error bars indicate SEM.

**Figure 4 biomolecules-11-01173-f004:**
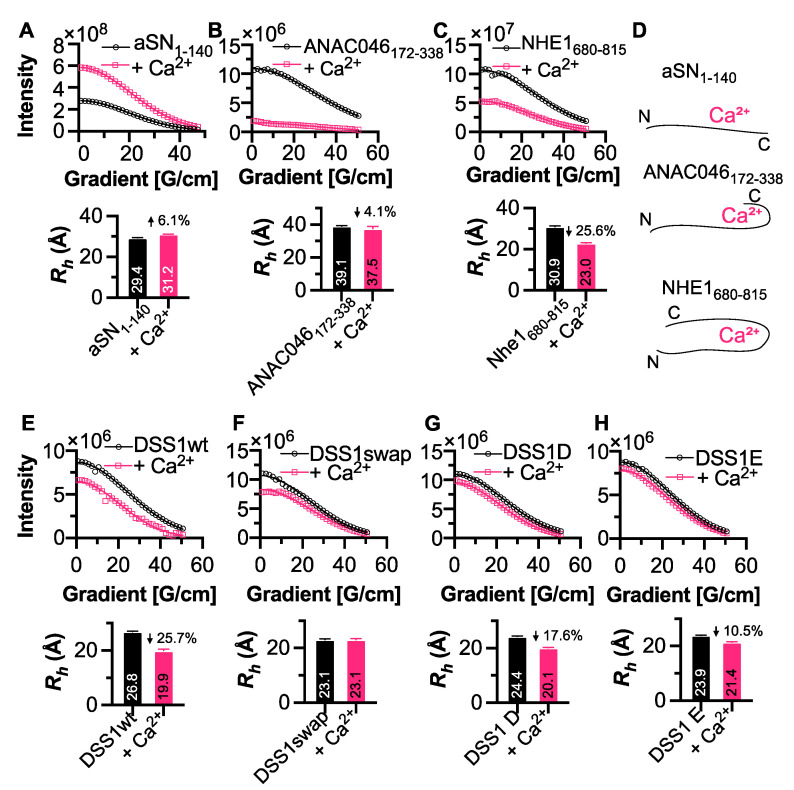
Calcium-induced conformational changes in IDPs. DOSY NMR data measuring the decay of the intensities of the methyl regions (centered within 0.85–0.9 ppm) as a function of increasing gradient strength. (**A**) DOSY NMR data demonstrating that CaCl_2_ causes an elongation of aSN_1-140_. (**B**) DOSY NMR data for ANAC046_172-338_ in the presence and absence of CaCl_2_. ANAC046_172-338_
*R_h_* was reduced by ~1.5 Å. (**C**) A conformational shift was observed in NHE1_680-815_, whereby the protein *R_h_* becomes ~8 Å smaller upon calcium binding. (**D**) Hypothetical conformational changes occurring upon CaCl_2_^+^ interaction. DOSY NMR data of DSS1wt (**E**), DSS1swap (**F**), DSS1D (**G**), and DSS1E (**H**). Arrows indicate increase (up) or decrease (down) in *R_h_*. All proteins were measured at a ratio of 1:250 to CaCl_2_.

**Table 1 biomolecules-11-01173-t001:** Properties of calcium-binding IDPs.

Protein	% D, E ^1^	% K, R ^1^	NCPR ^1^	κ ^1^	Hydropathy ^1^	pI	MW
							(kDa)
aSN_96-140_	33.3	6.7	−0.27	0.17	2.93	3.76	5.09
ANAC046_172-338_	10.1	4.2	−0.07	0.2	4.11	4.66	17.97
NHE1_680-815_	19.3	9.6	−0.1	0.27	3.65	4.38	14.51
DSS1wt	32.4	7	−0.25	0.39	3.53	3.67	8.1
ProTa	48.7	9	−0.4	0.42	2.53	3.66	12.2

^1^ Calculated using CIDER [23].

**Table 2 biomolecules-11-01173-t002:** Properties of Starmaker (SM) peptides.

Peptide	Sequence	% D, E ^1^	% K, R ^1^	NCPR ^1^	κ ^1^	Hydropathy ^1^	pI	MW (kDa)
SM1	VGTPAPADGDSVTTDGKDS	21.1	5.3	−0.16	0.22	3.76	3.77	1.79
SM2	DKPEGPDSDSAPDGDSASA	31.6	5.3	−0.26	0.16	3.01	3.61	1.82
SM3	ESVESKDHDSDSQDSDSAE	42.1	5.3	−0.37	0.16	2.53	3.80	2.07
SM4	HSDSDSDSHSDSDSDSDSD	42.1	0	−0.42	0.06	2.31	3.69	2.00

Bold indicates hydrophobic residues. ^1^ Calculated using CIDER [23].

## Data Availability

NMR assignments of ANAC046 are available at the BioMagResBank entry number 51033.

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
