# Peer review of "Insight into Calcium-Binding Motifs of Intrinsically Disordered Proteins"

_biomolecules, 2021, doi:10.3390/biom11081173_

Round 1

Reviewer 1 Report

This manuscript reported the screening of IDPs that bind to calcium and selected the various proteins and peptides; subsequently it was found that the selected proteins and peptides can be categorized into two groups. The categorized motifs were characterized in terms of charge propensity, charge distribution, charge localization, hydropathy, pI, length, and affinity towards Ca ion. NMR was used to obtain calcium-binding sites, Kd, and Rh, while colorimetric assay was used to obtain the amount of Ca ion bound by proteins and peptides. Generally, this manuscript is well written and organized, but I have several things which need to be addressed before acceptance for publication.

1) P3. L98: “C-terminal of the human” à “C-terminal region of the human”

2) P3. L103-106: The contents inside the parenthesis should be clarified.

3) P3. L114: “ANAC046172-338, DSS1” à “ANAC046172-338 and DSS1”

4) P5. L232: Is it adequate to assume the stoichiometry as 1-to-1? Could the authors rationalize this assumption?

For example, is the stoichiometry really 1-to-1 in the case of DSS1wt? Have authors tried Scatchared plot?

5) P6. L250: Please clarify the sentence, “This could be due to some of the … could not be followed.” Does it related to the pink triangles in Fig. 1(I)?

6) P6. L259-261: OCPC assay using 100 uM proteins is very interesting. It seems that DSS1wt and ProTα have multiple calcium binding sites and the stoichiometry of their binding seems not to be 1-to-1. Is it possible to obtain the Kd for each of the different binding sites, i.e. Kd for calcium 1, that for calcium 2, and so on, using NMR?

7) P8. Section 3.2: SM2 and SM4 exhibited binding with calcium. Do these peptides bind to calcium as a monomer? Does calcium induce oligomerization of the peptides? This point should be mentioned.

8) P9. L332: Are the Kd values in Figure 3 in mM?

9) P10. L348: “s” in “Kds” should not be subscript.

10) P10. L355: What does it mean by “… protein flexibility may help to account for the lack of glutamic acid.”? Please clarify.

11) P10. L361: The authors found conformational changes induced by calcium interaction. Does this conformational change also induce interprotein interactions, resulting in oligomerization? How does hydrophobicity of the proteins influence on this conformational change or get influenced by the conformational change?

12) P10. L381 Figure 4: Which signals did authors use to obtain the curves presented in the Figures? This information should be provided.

13) What would be the thermodynamics interpretation for the need of hydrophobicity in the IDRs for better binding with calcium? Discussion concerning this aspect would be very interesting.

14) Most importantly, authors should provide NMR spectra for all the proteins and peptides used in this study as supplementary information. The spectra should contain assignments; 2D 1H,15N-HSQC for proteins and finger printing regions of e.g. TOCSY for peptides. The original NMR spectra that are used for CSP analyses should also be provided.

Author Response

1) P3. L98: “C-terminal of the human” à “C-terminal region of the human”

This has been changed

2) P3. L103-106: The contents inside the parenthesis should be clarified.

We agree that the parenthesis was too congested and have now expanded it to be more readable.

3) P3. L114: “ANAC046172-338, DSS1” à “ANAC046172-338 and DSS1”

This has been changed

4) P5. L232: Is it adequate to assume the stoichiometry as 1-to-1? Could the authors rationalize this assumption?

For example, is the stoichiometry really 1-to-1 in the case of DSS1wt? Have authors tried Scatchared plot?

This is a general point brought up by all reviewers. We have therefore taken some time to perform new OCPC experiments that have allowed us to make Scatchard plots to calculate both stoichiometry and Kd. In doing so, we found that the Kd calculated by NMR was likely too high due to interference from ionic strength and the associated inductive effects on the chemical shifts. Thus, we have now excluded the titration data performed by NMR and instead incorporated the Scatchard plots as suggested by the reviewer into the manuscript. These can be found on pages 5-9 in the main text and as Figure S3. We thank the reviewer for this suggestion, as it has greatly improved the manuscript.

5) P6. L250: Please clarify the sentence, “This could be due to some of the … could not be followed.” Does it related to the pink triangles in Fig. 1(I)?

This is now explained in more details and explained in Fig. 1, please see the manuscript p. 6

6) P6. L259-261: OCPC assay using 100 uM proteins is very interesting. It seems that DSS1wt and ProTα have multiple calcium binding sites and the stoichiometry of their binding seems not to be 1-to-1. Is it possible to obtain the Kd for each of the different binding sites, i.e. Kd for calcium 1, that for calcium 2, and so on, using NMR?

Thank you for recognizing the value of the OCPC assay. We have now updated the manuscript to include data pertaining to the stoichiometry. Indeed, because we were working at concentrations above Kd, these data do indicate the stoichiometry. This value is also now extracted from Scatchard plot and we see good correspondence between these two different sets of data.

7) P8. Section 3.2: SM2 and SM4 exhibited binding with calcium. Do these peptides bind to calcium as a monomer? Does calcium induce oligomerization of the peptides? This point should be mentioned.

We have now included a sentence about the possibility of SM peptides to oligomerize through calcium-binding based on the stoichiometry of the interaction obtained from the Scatchard analyses. While we don’t think that SM2 or SM3 oligomerize, we have included “This indicates that two SM1 peptides may be required for an interaction with calcium and may hint that calcium induced oligomerization occurs.” (page 7).

8) P9. L332: Are the Kd values in Figure 3 in mM?

We have now removed this part of figure 3 (see comments to point 4) and inserted the new Scatchard plot analyses, however we have been careful to indicate the concentration on the graphs and in the figure legends.

9) P10. L348: “s” in “Kds” should not be subscript.

Thank you – this has been corrected

10) P10. L355: What does it mean by “… protein flexibility may help to account for the lack of glutamic acid.”? Please clarify.

Thank you for pointing this out, we have expanded on this in the revised manuscript (page 9).

11) P10. L361: The authors found conformational changes induced by calcium interaction. Does this conformational change also induce interprotein interactions, resulting in oligomerization? How does hydrophobicity of the proteins influence on this conformational change or get influenced by the conformational change?

This is an interesting point, however our Rh values do not suggest higher-order structures, although they may be present at very low populations. We are aware that other ion binding to IDPs can result in transient oligomers (Pedersen et al., Angew Chem, 2011) and more stable oligomers (Cragnell et al, Biomolecules, 2019), but in these cases there was good experimental evidence for their presence. Our data does not support their formation, and can likely be explained by the less stringent demands for coordination for calcium compared to zinc and copper. We think this is an important result to point out in the manuscript and have done so in the discussion on page 11. We thank the reviewer for this suggestion.

12) P10. L381 Figure 4: Which signals did authors use to obtain the curves presented in the Figures? This information should be provided.

Indeed, we are aware that for IDPs, the signals of the amides are not the most correct signals to follow for determination of diffusion due to the fast exchange with water, and therefore we measured the signal decay of the methyl region (outside the Dss signal). This information is important to include, and we thank the reviewer for pointing this out. The information is now included in the figure legend to Figure 4 and in the methods section.

13) What would be the thermodynamics interpretation for the need of hydrophobicity in the IDRs for better binding with calcium? Discussion concerning this aspect would be very interesting.

This is an interesting point, and possibly requires further experimental investigation. We have hypothesized that hydrophobic residues increase both entropy (from release of bound water) and enthalpy from van der Waals interactions) leading to the stabilization of the calcium ion in complex (page 11).

14) Most importantly, authors should provide NMR spectra for all the proteins and peptides used in this study as supplementary information. The spectra should contain assignments; 2D 1H,15N-HSQC for proteins and finger printing regions of e.g. TOCSY for peptides. The original NMR spectra that are used for CSP analyses should also be provided.

We have included the NMR spectra suggested in the supplementary file and refer to these in the main text on pages 7 and 8. The assignments have been included for the SM peptides, however, the protein assignments have been reported and can be found in the BMRB database. BMRB accession numbers can be found in the methods section on page 4.

Reviewer 2 Report

The material presented in the work is interesting and quite well prepared and presented, but I have a few comments:

  1. What is the Kd value is not clearly described.
  2. Why when calculating the constant Kd (regardless of whether it is a dissociation constant or a binding constant) it was assumed that the stoichiometry of the complexes is 1: 1? (It should also be noted that this fit assumes a 1: 1 protein to calcium binding - lines 232-233). The approximate stoichiometry of the complex is easy to read from OCPC measurements. I suggest recalculate Kd (whatever this value is) taking the correct, closer-to-truth assumptions about the complex stoichiometry.
  3. Why is the data shown without calcium ion concentration dependent CSPs see Figure 2 (Starmaker peptides)? I can only be suspicious but it seems that in this case the CSPs (at least for the data presented) are not a good measure of either the bond strength or the stability constant of the complexes. The greatest CSP changes are observed for the SM4 peptide, which binds the least amount of calcium ions per peptide molecule? Have you measured CSP values ​​for different concentrations of calcium ions. I understand that this is a naturally N15 peptide, however the changes in CSP shown are similar in amplitude to those measured for N15 labeled proteins, so why has this data not been recorded and shown?
  4. Again, it's about the Kd value, this time it's about the DSS1 protein. The data from the OCPC measurements show that the proteins DSS1wt, DSS1swap and DSS1D form complexes with a stoichiometry of 1: 4 (protein: calcium ion). I understand the results presented here (three Kd values ​​for three protein regions) that the protein only binds three calcium ions. Again, introduction the correct model for the complex stoichiometry should be a good ide. It is highly probable that assuming 1: 2 stoichiometry (protein: calcium ion) for one of the three regions, a better match will be obtained and the structure of the complex itself will be better understood (the sequence region binding two calcium ions will be defined).
  5. The studied proteins and peptides (especially the SM1 peptide) contain regions susceptible to isoasparagine formation, was the susceptibility of these sequences to this kind of chemical degradation under the conditions of the experiment tested?

Author Response

1. What is the Kd value is not clearly described.

We thank the reviewer for pointing this out. We have expanded on the experimental procedures and define here the dissociation constant Kd.

2. Why when calculating the constant Kd (regardless of whether it is a dissociation constant or a binding constant) it was assumed that the stoichiometry of the complexes is 1: 1? (It should also be noted that this fit assumes a 1: 1 protein to calcium binding - lines 232-233). The approximate stoichiometry of the complex is easy to read from OCPC measurements. I suggest recalculate Kd (whatever this value is) taking the correct, closer-to-truth assumptions about the complex stoichiometry.

This is an important point and we agree that this was not always clear in the manuscript. We have now revised our data and include new Scatchard plots obtained for all proteins and peptides. We were able to obtain both stoichiometry and Kd from these plots. In some cases, the interaction with calcium is 1:1, however we have found both ProTa and DSS1 interact with calcium at a higher stoichiometry. The data is now included in updated Figures 1, 2 and 3 as well as in the supplemental data.

3. Why is the data shown without calcium ion concentration dependent CSPs see Figure 2 (Starmaker peptides)? I can only be suspicious but it seems that in this case the CSPs (at least for the data presented) are not a good measure of either the bond strength or the stability constant of the complexes. The greatest CSP changes are observed for the SM4 peptide, which binds the least amount of calcium ions per peptide molecule? Have you measured CSP values ​​for different concentrations of calcium ions. I understand that this is a naturally N15 peptide, however the changes in CSP shown are similar in amplitude to those measured for N15 labeled proteins, so why has this data not been recorded and shown?

This is something we had considered, however it becomes a trade-off between being able to observe CSPs in the SM1 peptide and inducing large CSPs caused by a change in ionic strength, as is already visible for the SM4 peptide. To add to the data for the SM peptides, we have now included the Kd and stoichiometry (Figure 2F, G) as calculated from Scatchard plots (added as Figure S3). We have further elaborated on why we believe calcium induces CSPs in SM4 on page 7 as follows: “At a higher NCPR, as seen with SM4, CSPs do not necessarily reflect greater interaction, but rather the increased effect of ionic strength within the solution on the chemical shifts. It is therefore important to supplement NMR data with another assay, such as the OCPC assay, to be able to differentiate the effects of ionic strength from the interaction with calcium.” From these consideration we have reassessed whether NMR should be used for calculating Kd of proteins with calcium because of this issue with ionic strength. We thank the reviewer for spurring these changes.

5. Again, it's about the Kd value, this time it's about the DSS1 protein. The data from the OCPC measurements show that the proteins DSS1wt, DSS1swap and DSS1D form complexes with a stoichiometry of 1: 4 (protein: calcium ion). I understand the results presented here (three Kd values ​​for three protein regions) that the protein only binds three calcium ions. Again, introduction the correct model for the complex stoichiometry should be a good ide. It is highly probable that assuming 1: 2 stoichiometry (protein: calcium ion) for one of the three regions, a better match will be obtained and the structure of the complex itself will be better understood (the sequence region binding two calcium ions will be defined).

We agree that the OCPC data is a powerful tool for understanding the interaction, so we expanded on the assay by new titrations that allowed for generating Scatchard plots to determine both stoichiometry and Kd. In doing so, we found issues with the use of NMR for measuring Kd in this particular case as the increased ionic strength induced by the addition of calcium chloride also affect the chemical shifts. Thus, we have been able to obtain more reliable Kds and thus calculate the stoichiometry more accurately from these plots.

5. The studied proteins and peptides (especially the SM1 peptide) contain regions susceptible to isoasparagine formation, was the susceptibility of these sequences to this kind of chemical degradation under the conditions of the experiment tested

From the NMR assignments the sequences were all intact and no isoasparagine formation was detected in any of the proteins and peptides. See also the new supplementary figure S3, where we show the assignments of the Starmaker peptides.

Reviewer 3 Report

Newcombe et al. investigated calcium-binding motifs in IDPs using solution NMR spectroscopy. It is important to understand the biological roles of calcium binding to IDPs. The authors found at least two groups of calcium-binding sequences, an Excalibur-like motif similar to the EF-hand loop and a condensed-charge motif carrying negative charges. The experimental results indicate that negative charges and sufficient hydropathy in combination are required of IDPs to interact with calcium. Most of conclusions are justified by experimental results. However, some points need to be improved before acceptance.

  • Page 5, line 232: The chemical shift data were analyzed assuming a 1:1 protein to calcium binding. It should be explained why the authors assumed a 1: 1 bond. Indeed, DSS1wt seems to have multiple calcium binding sites (page 9, 344).
  • Figure 2: Although larger CSPs were observed in SM4 than others, it bound significantly less calcium. How can you explain the CSPs? It is better to have brief discussion in the manuscript.
  • I guess that CSP is a combination of 1H and 15N(or 13C) chemical shifts. It should be mentioned how to weight the relative chemical shifts of the different nuclei.
  • 5℃ -- > 5 ℃ (”o” is not unit, but “℃” is unit.)

Author Response

  • Page 5, line 232: The chemical shift data were analyzed assuming a 1:1 protein to calcium binding. It should be explained why the authors assumed a 1: 1 bond. Indeed, DSS1wt seems to have multiple calcium binding sites (page 9, 344).

Spurred by the comments from reviewer 1 and 2, we have extracted Kds using the Scatchard plots. We have therefore removed this part of the study and use the NMR titrations to pinpoint the interactions sites and the Scatchard plots to extract Kds and stoichiometry.

Figure 2: Although larger CSPs were observed in SM4 than others, it bound significantly less calcium. How can you explain the CSPs? It is better to have brief discussion in the manuscript.

This is an interesting question that was also also asked by reviewer 2, and we have added text to the manuscript to clarify. Indeed, this is something we had considered, but it becomes a trade-off between being able to observe CSPs in the SM1 peptide and inducing large CSPs caused by a change in ionic strength, as is already visible for the SM4 peptide. To add to the data for the SM peptides, we have now included the Kd and stoichiometry (Figure 2F, G) as calculated from Scatchard plots (added as Figure S3). We have further elaborated on why we believe calcium induces CSPs in SM4 on page 7 as follows: “At a higher NCPR, as seen with SM4, CSPs do not necessarily reflect greater interaction, but rather the increased effect of ionic strength within the solution on the chemical shifts. It is therefore important to supplement NMR data with another assay, such as the OCPC assay, to be able to differentiate the effects of ionic strength from the interaction with calcium.” We thank the reviewer for spurring these changes.

  • I guess that CSP is a combination of 1H and 15N(or 13C) chemical shifts. It should be mentioned how to weight the relative chemical shifts of the different nuclei.

The CSP are derived from the amide shifts and this is now clarified in the manuscript p. 4.

  • 5℃ -- > 5 ℃ (”o” is not unit, but “℃” is unit.

This is now corrected.